# Microanatomical organization of hepatic venous lymphatic system in humans

**Kotaro Umemura**[1,2], **Hiroshi Shimoda**[2,3], **Keinosuke Ishido**[1], **Norihisa Kimura**[1], **Taiichi Wakiya**[1], **Takuji Kagiya**[1,2], **Kentaro Sato**[1,2], **Yuto Mitsuhashi**[1,2], **Seiji Watanabe**[2], **Hirokazu Narita**[4], **Tomohiro Chiba**[2], **Kenichi Hakamada**[1]*

1 Department of Gastroenterological Surgery, Hirosaki University, Graduate School of Medicine, Hirosaki, Aomori, Japan, 2 Department of Anatomical Science, Hirosaki University, Graduate School of Medicine, Hirosaki, Aomori, Japan, 3 Department of Neuroanatomy, Cell Biology, Histology, Hirosaki University, Graduate School of Medicine, Hirosaki, Aomori, Japan, 4 Biomedical Science and Engineering Research Center, Hakodate Medical Association Nursing and Rehabilitation Academy, Hakodate, Hokkaido, Japan

* hakamada@hirosaki-u.ac.jp

**Data Availability Statement:** All relevant data are within the manuscript.

## Abstract

Lymphatic fluid drains from the liver via the periportal lymphatic, hepatic venous lymphatic, and superficial lymphatic systems. We performed a postmortem study to clarify the three-dimensional structure and flow dynamics of the human hepatic venous lymphatic system, as it still remains unclear. Livers were excised whole from three human cadavers, injected with India ink, and sliced into 1-cm sections from which veins were harvested. The distribution of lymphatic vessels was observed in 5 μm sections immunostained for lymphatic and vascular markers (podoplanin and CD31, respectively) using light microscopy. Continuity and density of lymphatic vessel distribution were assessed in en-face whole-mount preparations of veins using stereomicroscopy. The structure of the external hepatic vein wall was assessed with scanning electron microscopy (SEM). The lymphatic dynamics study suggested that lymphatic fluid flows through an extravascular pathway around the central and sublobular veins. A lymphatic vessel network originates in the wall of sublobular veins, with a diameter greater than 110 μm, and the peripheral portions of hepatic veins and continues to the inferior vena cava. The density distribution of lymphatic vessels is smallest in the peripheral portion of the hepatic vein (0.03%) and increases to the proximal portion (0.22%, p = 0.012) and the main trunk (1.01%, p < 0.001), correlating positively with increasing hepatic vein diameter (Rs = 0.67, p < 0.001). We revealed the three-dimensional structure of the human hepatic venous lymphatic system. The results could improve the understanding of lymphatic physiology and liver pathology.

## Introduction

The liver produces a large amount of lymphatic fluid, of which 25%–50% flows through the thoracic duct [1]. Lymphatic fluid produced from the intrahepatic portal vein and intrahepatic artery drains through three pathways: the periportal lymphatic system along the portal triad; the hepatic venous lymphatic system along the hepatic veins; and the superficial lymphatic

**Funding:** The authors received no specific funding for this work.

**Competing interests:** The authors have declared that no competing interests exist.

system of the liver capsule [2–9]. The hepatic venous lymphatic system has been studied in several mammals [10, 11]. Ohtani et al. observed collagen fiber bundles around central and sublobular veins on electron microscopy and suggested that they might work as an extravascular fluid pathway [12]. Poonkhum et al. performed a microanatomical study of the hepatic venous lymphatic systems of cats; they ligated the thoracic duct, which led to the dilation of the sinusoids, the space of Disse, interstitial space of sublobular veins, and sublobular lymphatic vessels [10]. The presence of immune cells in these dilated areas indicated that lymphatic fluid produced in the liver enters sublobular lymphatic vessels from the space of Disse through the sublobular interstitial space [10]. Although the findings so far suggest that the hepatic venous lymphatic system involves a drainage pathway along hepatic veins, the details on its structure and function in the human liver have not been elucidated. In this study, we performed a microanatomical analysis of the human liver to reveal the three-dimensional structure and lymphatic flow dynamics of the hepatic venous lymphatic system.

## Materials and methods

### Study design

This was a postmortem liver study. Specimens of human liver were collected from three humans (Table 1). Donor was enrolled before his death in an association of people who offered to donate their bodies with the consent of themselves and their relatives, provided that they were not infected with human immunodeficiency virus, hepatitis B virus, or hepatitis C virus. Additionally, we obtained consent from the relatives posthumously. The study was approved by the Human Research Ethics Committee of Hirosaki University Graduate School of Medicine (reference number 2019–1075). We received opt-out consent from the relatives for this study.

### The details of cadavers

Postmortem, 10% formalin and 5% phenol were injected through the femoral artery of the cadavers and fixed. They had no hepatic tumors or abdominal malformations. The details of the cadavers used in the experiment were shown in Table 1. The main studies on the three cadavers are as follows. Cadaver No.1: Sections across the hepatic veins and inferior vena cava were prepared and studied to clarify the lymphatic anatomy around the hepatic veins, including the sublobular vein. Cadaver No.2: To elucidate the lymph dynamics around the hepatic veins, sections were prepared from liver specimens in which Indian ink was injected into the right lobe and examined using microscope. Cadaver No.3: To clarify the three-dimensional structure and distribution of lymphatic vessels around hepatic veins, whole mount immunostaining was performed on a block of hepatic vein with the liver parenchyma removed and examined using stereomicroscope.

**Table 1. Details of the three cadavers.**

| No. | Age | Sex | Cause of death | Preservation time until specimen collection | Pathological findings of the liver | Goal of the experiment |
|-----|-----|-----|----------------|---------------------------------------------|------------------------------------|------------------------|
| 1 | 75 | male | suffocation | 22 months | mild hepatitis | Lymphatic distribution around hepatic veins |
| 2 | 88 | female | cerebral hemorrhage | 15 months | moderate hepatitis | Lymphatic dynamics around hepatic veins |
| 3 | 91 | female | acute pneumonia | 22 months | almost normal | The three-dimensional structure of lymphatic vessels around hepatic veins |

## Specimen collection

The livers were excised whole; one of them was injected with 5 mL India ink (into the right lobe parenchyma) to investigate the flow of interstitial tissue fluid. Each liver was sectioned coronally approximately every 1 cm. Several hepatic veins were harvested from each hepatic segment. After removal of as many hepatic plates as possible, samples were embedded in paraffin and sliced to approximately 5 μm thickness for use in SEM.

## Light microscopy

The 5 μm tissue sections were deparaffinized and stained with hematoxylin and eosin. They were then incubated in 0.01 M citrate buffer (pH 6.0) at 121°C for 15 min to retrieve the antigenicity of the target proteins before immunostaining. The sections were immersed in 0.3% $H_2O_2$ in PBS (1/15 M, pH 7.4) for 20 min at room temperature to block endogenous peroxidase activity. After washing with 1/12 M PBS (pH 7.4), they were incubated in 5% normal goat serum (Vector Laboratories, Burlingame, CA, USA) for 15 min, followed by incubation with a mixture of antibodies against the lymphatic marker podoplanin (D2-40, DAKO, Santa Clara, CA, USA; ×100 dilution) and the vascular marker CD31 (EP3095; ×250 dilution, Abcam, Tokyo, Japan) at 4°C overnight. Following rinsing in PBS, the sections were treated with anti-mouse IgG conjugated to alkaline phosphatase (AP; Histofine Simple Stain AP, Nichirei Biosciences, Tokyo, Japan) for 2 h at room temperature, and the immunoreaction was visualized with the AP reaction Vector Blue Substrate Kit (Vector Laboratories). They were then treated with anti-rabbit IgG conjugated to horseradish peroxidase (Histofine Simple Stain MAX-PO, Nichirei Biosciences) followed by visualization of the immunoreaction by use of diaminobenzidine (DAB; DAKO). The stained sections were examined using a BX-50 light microscope equipped with a DP72 digital imaging system (Olympus, Tokyo, Japan). The wall thickness of the sublobular vein with lymphatic vessels was measured using ImageJ software.

## Stereomicroscopy of lymphatic vessel continuity and distribution

The harvested hepatic vein samples were readied for en-face whole-mount preparation by rinsing in PBS containing 3% Triton X-100 (FUJIFILM Wako Pure Chemical Corporation, Osaka, Japan) and were immersed in 5% normal goat serum (Vector Laboratories) for 2 days at 4°C, then incubated with an antibody to podoplanin (DAKO) for 14 days at 4°C while being gently shaken. After washing with PBS with Triton, immunoreaction was shown with anti-mouse IgG conjugated to AP (Nichirei Biosciences) and AP reaction medium [13]. The vessels were cut longitudinally and mounted on glass slides.

Stereomicroscopy was performed with an MVX100 stereomicroscope and recorded with the DP80 imaging system (Olympus). The hepatic veins were divided into three portions under the stereomicroscope and classified by size as follows: the main trunk (large vessels draining to the inferior vena cava), the peripheral portion (slender vessels receiving numerous sublobular veins), and the proximal portion (intermediate veins bridging between the main trunk and peripheral portions). Continuity across the different proportions was assessed.

For quantification of the lymphatic vessels, the tissues were processed with a tissue-clearing treatment using the reagents CUBIC-HL and CUBIC-R+ (Tokyo Chemical Industry, Tokyo, Japan) in accordance with the manufacturer's protocol. The cleared tissues were viewed under a stereomicroscope and the ratio of the area positive for podoplanin to the total vein wall area was calculated after processing the images with ImageJ software. If the area of a specimen was large and required multiple images from different parts of the sample, all the images were merged in Adobe Photoshop CC (Adobe Systems, San Jose, CA, USA) before calculation of the lymphatic area. The diameter of each venous segment was also measured in the processed

images and the correlation between vessel size and density of lymphatic vessel distribution was assessed. The estimated number of samples of the peripheral portion, the proximal portion, and the main trunk were 30, 29, and 14, respectively.

### Scanning Electron Microscopy (SEM)

Specimens containing peripheral venous segments were processed for SEM as described previously [14] to examine the outer wall of hepatic veins in which lymphatic vessels had been observed in the adventitia. Briefly, the tissues were stained en bloc with 1% osmium tetroxide solution and 1% tannic acid for 1 h each, and then postfixed in 1% osmium solution for 1 h. They were dehydrated in a graded ethanol series, freeze-dried with t-butyl alcohol, and observed under a JEOL JSM-6510 (JEOL, Tokyo, Japan).

### Schematic illustration of the hepatic venous lymphatic system

The lymphatic network around hepatic veins was illustrated as a model that was reconstructed from data obtained using light microscopy, stereomicroscopy, SEM, and statistical analysis. The schematic illustration was created with Adobe Illustrator CC (Adobe).

### Statistical analysis

Comparisons of the density of lymphatic vessels in different portions of the hepatic vessels were made using the Kruskal–Wallis test. We further used the Spearman rank correlation coefficient to evaluate the associations between the diameters of the hepatic vein segments and the density distribution of lymphatic vessels. The samples of peripheral portions were excluded because there were few lymphatic vessels in many samples. Two-tailed values of $p < 0.05$ were regarded as significant in each analysis. All statistical analyses were performed using EZR (Saitama Medical Center, Jichi Medical University, Japan), a graphical user interface for the R software program (The R Foundation for Statistical Computing, Vienna, Austria).

## Results

### Lymphatic vessel locations

Lymphatic vessels were observed throughout the adventitia and intima of the inferior vena cava but were mainly in the adventitia in hepatic veins (Fig 1A and 1B). In addition, a small number of lymphatic vessels were found in the adventitia of some sublobular veins (Fig 1C) that had a wall thickness greater than 110 μm. Lymphatic vessels were not found in central veins or in sublobular veins with walls thinner than 100 μm (Fig 1D).

### Lymphatic vessel continuity

Continuity was seen from lymphatic vessels in the hepatic vein peripheral portion through to the proximal portion and the main trunk into the inferior vena cava (Fig 2A–2D). Lymphatic vessels with blind ends were also seen in the peripheral portion, and no lymphatic vessels were observed in the peripheral hepatic veins. Blind ends were also observed in the proximal portion and the main trunk of hepatic veins (Fig 2B–2D).

### Distribution of lymphatic vessels in the hepatic veins

Microscopically, dense lymphatic vessels were observed in the more central part of the hepatic veins. Therefore, we examined the relationship between the diameter of hepatic veins and the density of lymphatic vessels. The ratio of lymphatic vessels to the vessel area differed

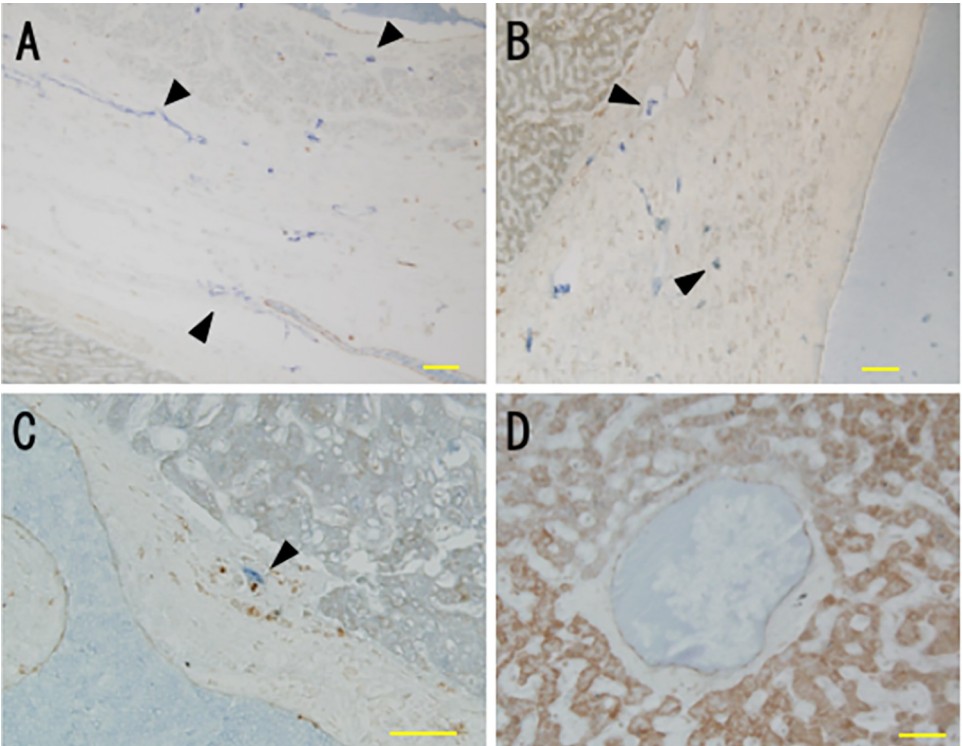

**Fig 1. Distribution of lymphatic vessels in hepatic veins.** Vessels were stained for podoplanin and CD31 in 5 μm sections and were assessed by light microscopy. (A) Inferior vena cava. (B) Hepatic vein. (C) Sublobular vein. (D) Central vein. Lymphatic vessels (arrowheads) were present in the adventitia and intima of the inferior vena cava, in the adventitia of hepatic veins and in some sublobular veins that had a wall thickness greater than 110 μm. Lymphatic vessels were not observed in central veins or most other sublobular veins. Scale bars: 100 μm (A, B); 50 μm (C, D).

significantly between the different portions of the hepatic veins (peripheral portion, 0.03%; proximal portion, 0.22%; and main trunk, 1.01%; Fig 3). We also showed that there was a direct correlation between the distribution rate of lymphatic vessels and the diameter of the hepatic vein (Fig 4), supported by more abundant lymphatic vessels in areas where hepatic veins were more centrally located (Fig 2B).

## Lymphatic fluid flow around hepatic veins

We observed India ink in the perivascular space and interstitium around central veins and small sublobular veins. India ink was consistently seen in vein walls in all portions of the hepatic vein and the inferior vena cava (Fig 5A–5E). India ink was also observed around the Gleason's sheath and diffusely around the sinusoids (Fig 5F).

## External structure of hepatic veins on SEM

In specimens where lymphatic vessels were observed, we found that the outer walls of hepatic veins were mainly composed of collagen fibrils, having a cribriform structure with small pores of about 10 μm in size (Fig 6A). In the walls of sublobular veins, we observed a meshwork structure consisting of collagen fibrils containing lymphocytes (Fig 6B and 6C).

## Discussion

Liver tumors, including metastatic liver tumors and primary liver cancer, are known to metastasize to mediastinal lymph nodes. The prognosis for patients with mediastinal lymph node

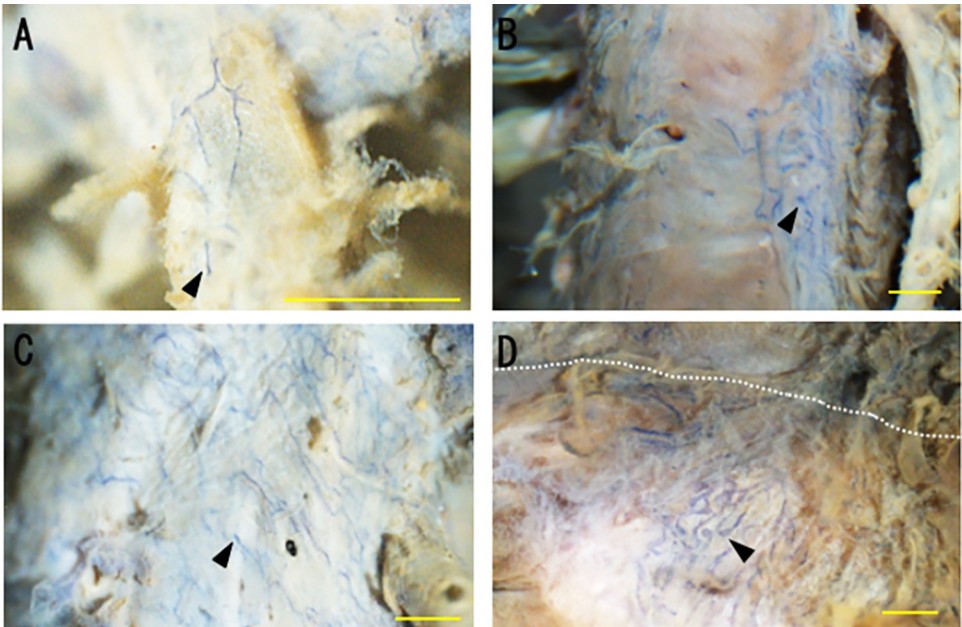

**Fig 2. Continuity and density distribution of lymphatic vessels from hepatic veins to the inferior vena cava.**
Vessels were stained for podoplanin and CD31 and assessed as en-face whole-mount preparations by
stereomicroscopy. (A) Peripheral portion: lymphatic vessels with blind ends were observed (arrowhead). (B) Proximal
portion: lymphatic vessels were mainly continuous and became more abundant in the area where the hepatic veins
were more centrally located. (C) Main trunk of the hepatic vein: lymphatic vessels were observed all around the wall of
the main trunk. (D) Border between the inferior vena cava and the right hepatic vein: the lymphatic vessels of the main
trunk of the hepatic vein were continuous with those of the inferior vena cava (the border is indicated by a perforated
line), forming a rich lymphovascular network. The blind ends of lymphatic vessels were also observed in the proximal
portion and the main trunk of the hepatic vein (B-D). Scale bars: 1 mm.

metastases is extremely poor, and little is known about the pathway of metastasis [15, 16].
Therefore, clarification of intrahepatic lymphatic anatomy is needed to elucidate the pathway
of metastasis. We focused on the lymphatic anatomy around the hepatic veins.

Using three different microscopic techniques, we revealed the three-dimensional structures
of the hepatic venous lymphovascular networks in human livers. No lymphatic vessels are seen
in the central vein or most sublobular veins. Instead, lymphatic fluid flows along an extravas-
cular pathway in the perivascular space and interstitium. The lymphatic vessel network origi-
nates mainly in the peripheral portion of the hepatic veins where they meet many sublobular
hepatic veins. Lymphatic fluid enters veins through pores in the cribriform outer walls and
flows into the lymphatic vessels via blind ends. The distribution of lymphatic vessels becomes
increasingly dense in the proximal portions and the main trunks of hepatic veins, creating a
rich lymphovascular network in the inferior vena cava (Fig 7).

The lymphatic vessels of hepatic veins have been reported to originate from the venous wall
of sublobular veins [10, 11] and were found to continue macroscopically to five or six separate
lymphatic vessels of the inferior vena cava [3, 17]. However, little information has been pro-
vided regarding the continuity and distribution of these lymphatic vessels. Our study revealed
that lymphatic vessels are found in the tunica adventitia of sublobular veins with a wall thick-
ness greater than 110 μm. Thus, more lymphatic vessels seem to be present in thinner sublobu-
lar veins than suggested by the assessment of cat livers [10]. This may suggest that the
lymphatic vessels around the hepatic veins originate from the adventitia of the sublobular
veins and develop more densely from the hepatic veins to the inferior vena cava (Fig 7).

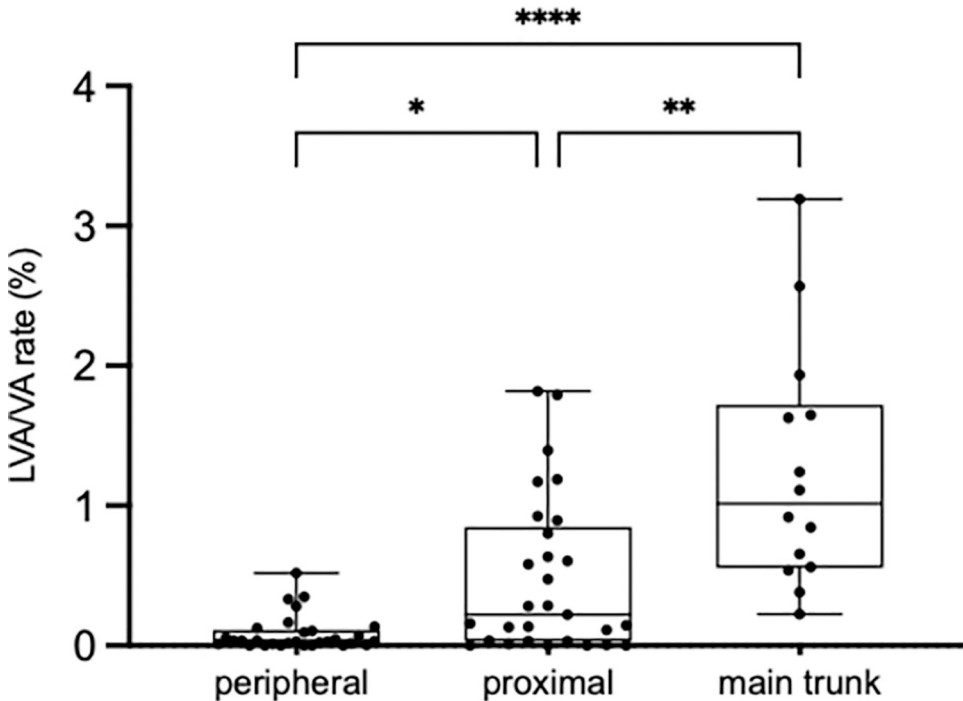

**Fig 3. Distribution rate of lymphatic vessels in each portion of the hepatic vein.** The distribution ratio of lymphatic vessels area was defined as the ratio of the podoplanin-positive area to total vein wall area. The distribution ratio increased from the peripheral portion (0.03%) to the proximal portion (0.22%) and to the main trunk (1.01%). *p = 0.012. **p = 0.0074. ****p<0.0001. Abbreviation: LVA/VA = ratio of lymphatic vessels to total area of hepatic vein wall.

In our study, lymphatic vessels were consistently observed from the peripheral and proximal portions to the main trunk of the hepatic vein. Although continuity has previously been reported between lymphatic vessels in sublobular veins, hepatic veins, the inferior vena cava, and the diaphragm [3, 17, 18], most previous observations have been made by macroscopic observation, light microscopy, or electron microscopy of tissue sections. Therefore, it was difficult to evaluate the three-dimensional structure of lymphatic vessels of hepatic veins [19]. The use of whole-mount immunohistochemistry stereomicroscopy of hepatic veins enabled us to evaluate the distribution of lymphatic vessels and their continuity.

The blind ends of the lymphatic vessels in the liver had a button-like appearance, similar to the blind ends of lymphatic vessels observed in other organs [20, 21]. There is a relatively loose arrangement of lymphatic endothelial cells, which plays an important role in the transport of lymphocytes and lymphatic fluid from the connective tissue of the veins into the lymphatic vessels [22, 23]. We found blind ends of the lymphatic vessels in all portions of hepatic veins. This finding suggests that immune cells and interstitial fluid enter the lymphatic vessels through the blind ends.

In this study, the density distribution of lymphatic vessels increased as veins became more centrally located. These results suggest that intrahepatic lymphatic vessels develop in the direction of the blood flow as in hepatic veins. The development of lymphatic vessels contributes to efficient lymphatic fluid transport. There have been no reports on the density of distribution of lymphatic vessels along hepatic veins in humans. Magari et al. reported that perivascular collagen fibrils in rabbit livers increased in proportion from central veins toward sublobular veins [18]. They mentioned that lymphatic flow through collagen fibers depends on blood

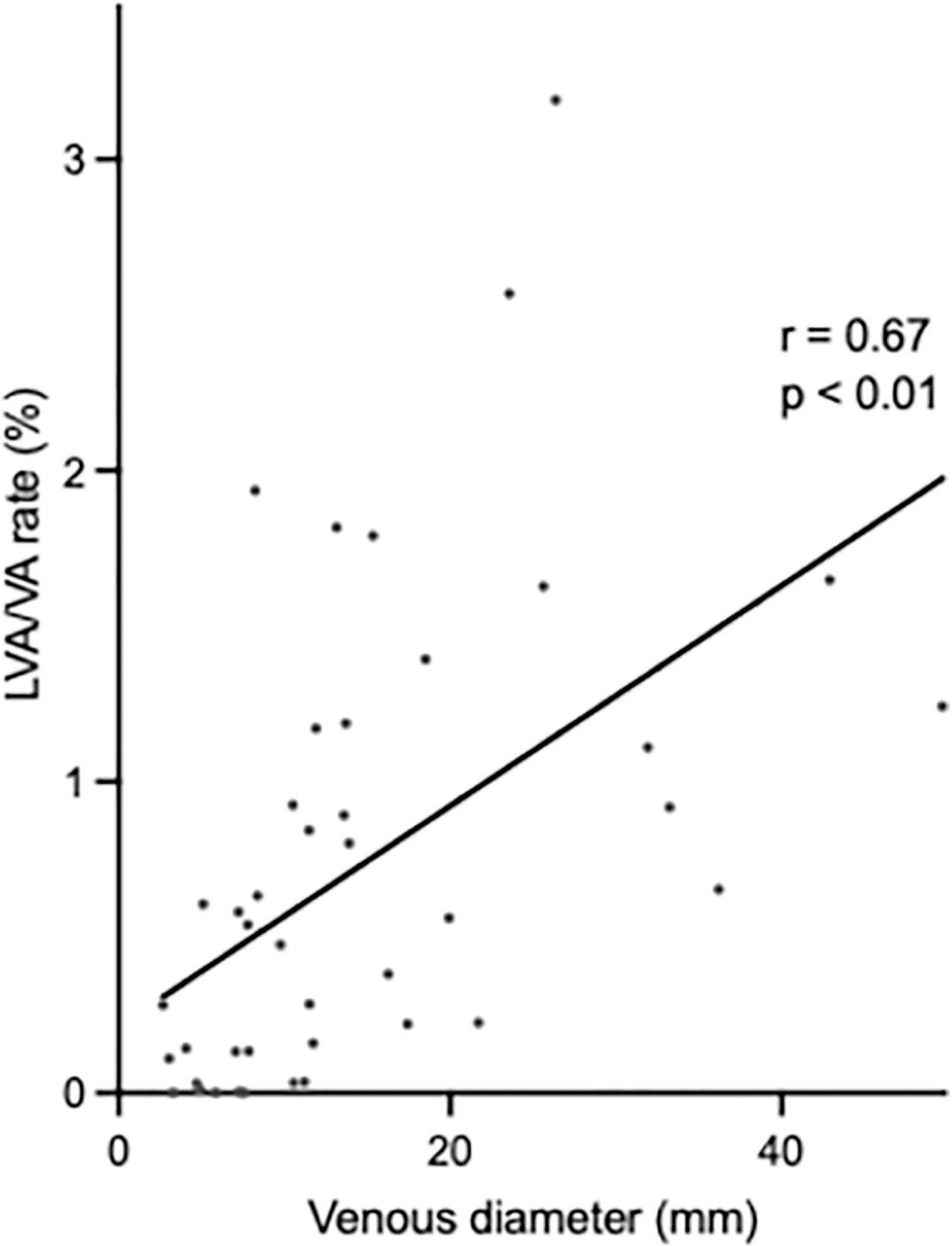

**Fig 4. Correlation between density distribution of lymphatic vessels and hepatic vein diameter.** Correlations were calculated using proximal and main trunk portions only, as few lymphatic vessels were present in the peripheral portions. The Spearmann's rank correlation coefficient between the diameter of the hepatic veins and the distribution rate of lymphatic vessels (LVA/VA ratio) was calculated. The distribution ratio of lymphatic vessels increased significantly as the diameter of the hepatic veins became larger. (Rs = 0.67, p<0.001). Abbreviation: LVA/VA = ratio of lymphatic vessels to total area of hepatic vein wall.

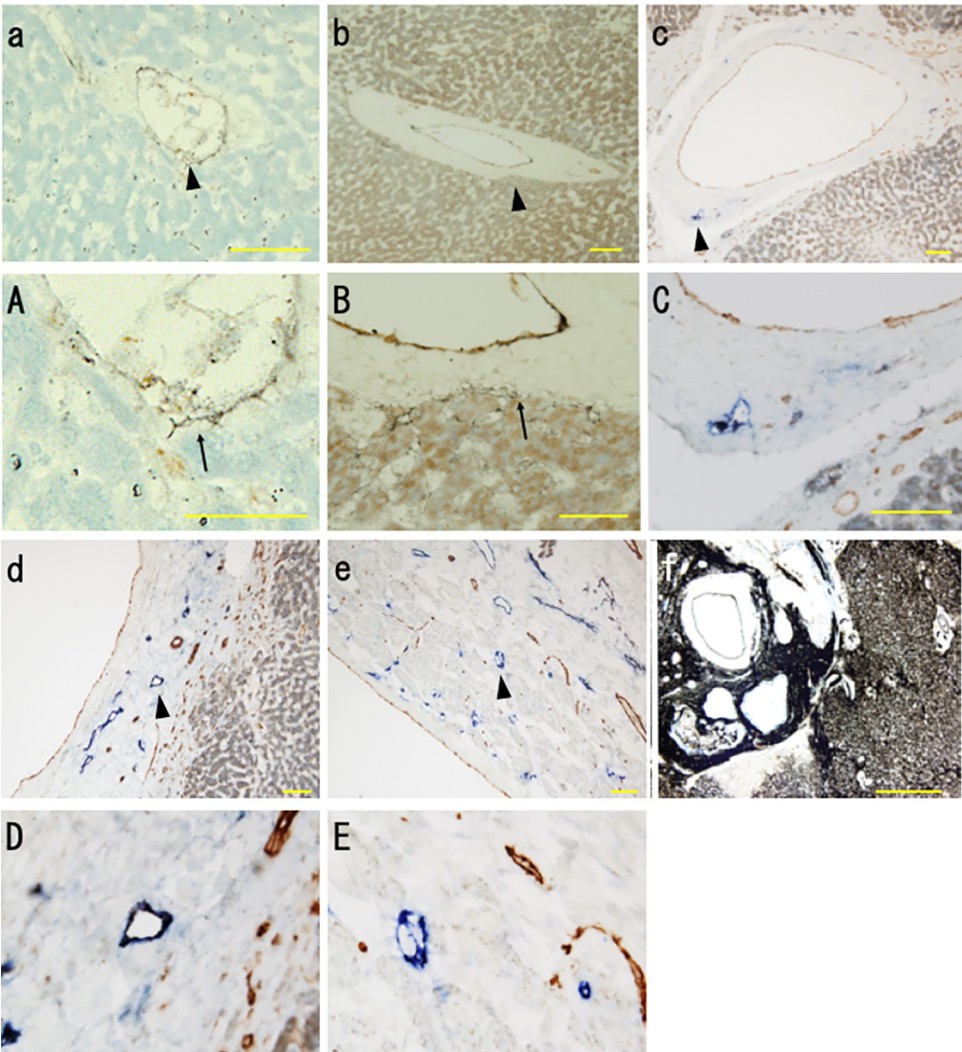

**Fig 5. The distribution of lymphatic vessels and India ink in the hepatic veins (Immunohistochemistry for podoplanin and CD31).** The uppercase letters show a magnified view of the area of the arrowhead in lowercase. a, A; central vein: India ink was observed in the perivascular space and in the interstitium of the central vein wall (arrow) b, B; sublobular vein: India ink was observed in the perivascular space and in the interstitium of the sublobular vein wall (arrow). c, C; sublobular vein with lymphatic vessel: India ink was present in the lymphatic vessels (arrowhead) of the sublobular veins. Scale bars: 50 μm (b, B), 100 μm (a, A, c, C,). d, D: hepatic vein. e, E; Vena Cava inferior. India ink was also present in the lymphatic vessels (arrowhead) of the hepatic veins and Inferior vena cava. f; Indian ink injection point: India ink injected into the liver parenchyma flowed into sinusoids and the Gleason's sheath. Scale bars: 100 μm (d, D, e, E), 500 μm (f).

pressure and permeability. In addition, Ohtani et al. also suggested that collagen fibrils of the hepatic veins functioned as one of the lymphatic fluid pathways [24]. These studies support our assessment of the ratio of lymphatic vessels to vessel area. Lymphatic vessels are thought to allow more efficient lymphatic transport than collagen fibers.

Ohtani et al. examined lymphatic dynamics in the liver using horseradish peroxidase as a tracer and found that HRP injected into the systemic circulation flowed into the periportal lymphatic system [25]. They also showed that the collagen fibrils in the sinusoids continued along with those of central and sublobular veins and functioned as a fluid pathway [24–26]. India ink, as used in this study, is a widely used dye for research of lymphatic vessels [27, 28].

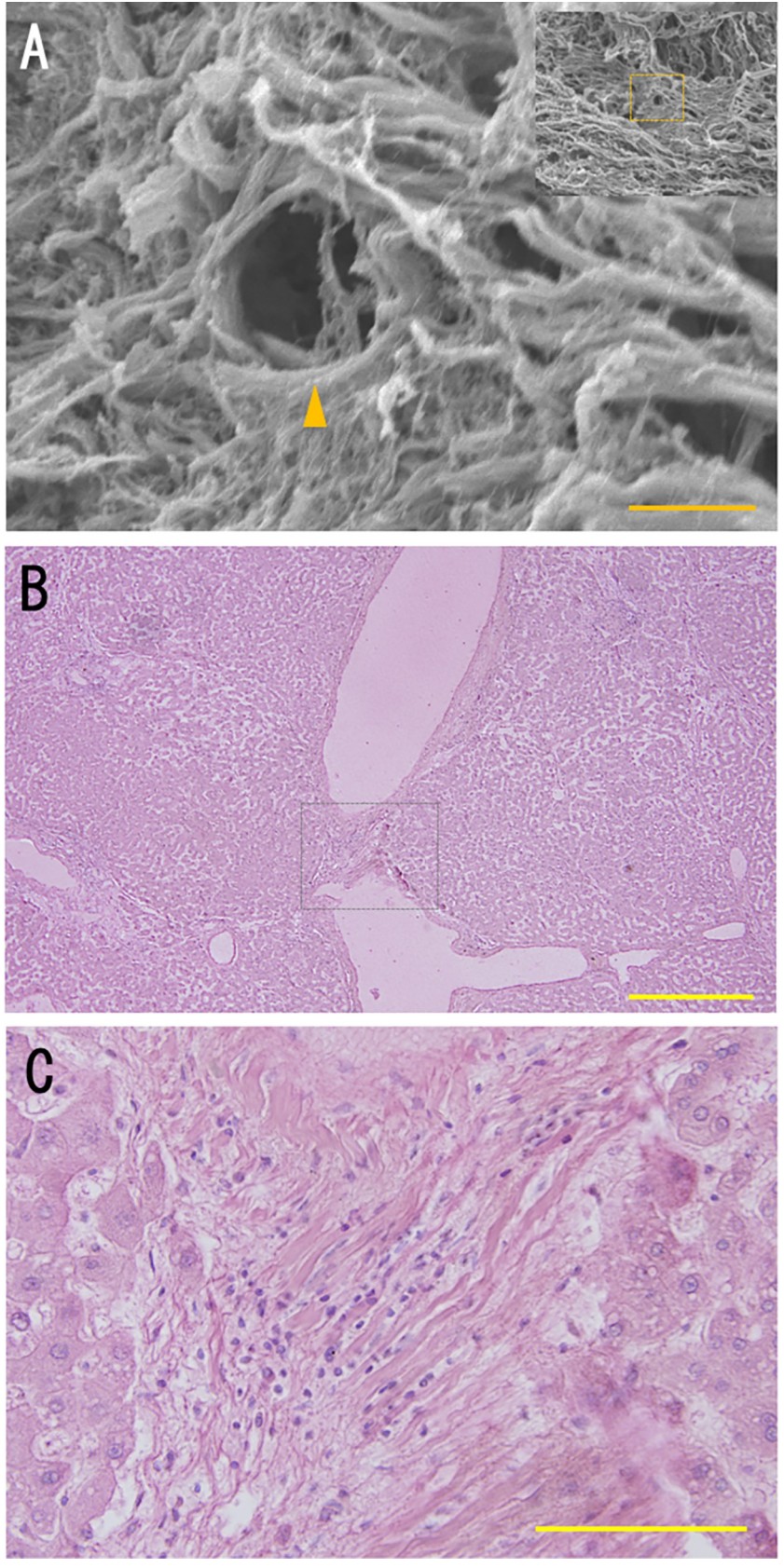

**Fig 6.**

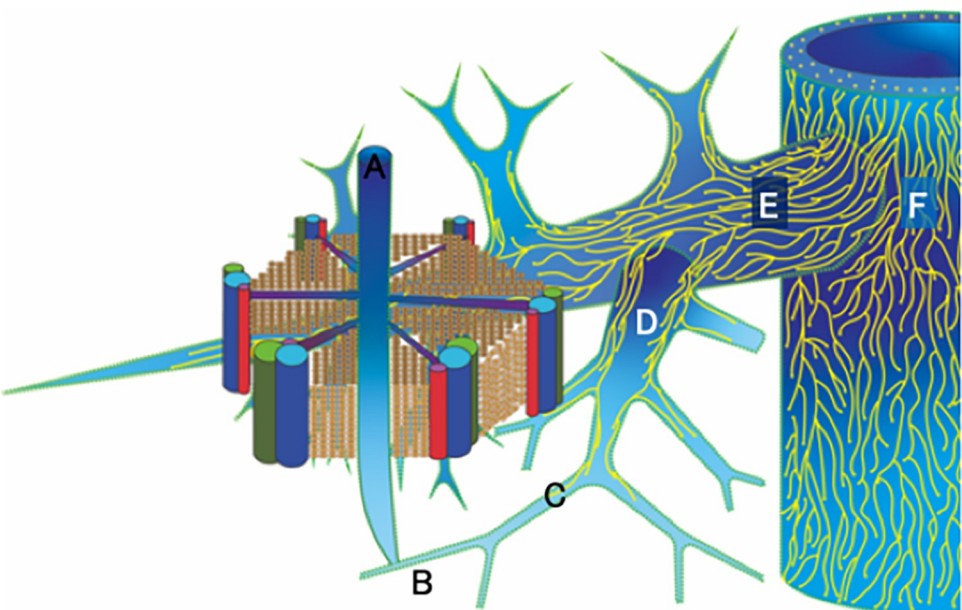

**Fig 7. Schema of lymphovascular networks from the central vein to the inferior vena cava.** A: central vein B: sublobular vein C: peripheral portion D: proximal portion E: main trunk F: inferior vena cava. Some of the lymphatic fluid derived from the space of Disse flows towards the central vein (A). The lymphatic fluid runs through the perivascular space (represented by green perforated lines) of the central vein and sublobular vein as an extravascular fluid pathway. Secondarily, it enters the interstitium of the vein through the small pores of the cribriform structure of the veins. And then, lymphatic fluid enters the lymphatic vessels of the peripheral portion of the hepatic veins (C). Most of the lymphatic fluid flows through the lymphatic vessels to the inferior vena cava (F). As the hepatic veins become thicker and more centrally located, the lymphatic vessels form a more intimate network of lymphovascular system.

After injection into the liver parenchyma, we observed India ink in the perivascular space of central veins and the walls of sublobular veins, whereas in larger sublobular veins, the hepatic vein, and inferior vena cava, the dye was seen in the lymphatic vessels of these walls. The small pores we found in the cribriform structure on the outer wall of the sublobular vein suggested a lymphatic pathway from the perivascular space into the vein wall. These pores were similar in appearance to the milky spots found on the greater omentum that function as a pathway for immune cells and tumor cells to lymphatic vessels [29]. Furthermore, lymphocytes were observed in the meshwork structure consisting of collagen fibrils in the wall of sublobular veins, supporting initial transport through an extravascular pathway in the collagen meshwork structure of the venous wall, before migrating into lymphatic vessels.

Frenkel et al. reported the drainage patterns involving a pathway from the inferior vena cava to mediastinal lymph nodes in the murine livers in vivo [30]. They showed that much of the lymphatic fluid from the middle lobe of the liver flows to the peri-esophageal lymph nodes via the lymphatic vessels around the IVC. They also showed that lymphatic fluid from the right and left lobes of the liver flows to the hilar lymph node. There may be heterogeneity in lymphatic drainage patterns for each region in human liver.

The lymphatic flow pathways were considered as follows: (i) some of the lymphatics of the space of Disse flow to the perivascular space of the central and sublobular veins. (ii)The lymphatics flow into the collagen fibrils of the veins via the small pores of the vein walls; (iii) they then enter the blind end and flow through the lymphatic vessels.

Patients with hepatic malignancies, such as hepatocellular carcinoma, intrahepatic cholangiocarcinoma, and metastatic liver tumors, are often diagnosed as having mediastinal lymph

node metastasis, for which the prognosis is extremely poor [3, 31, 32]. Although the superficial lymphatic system of the liver is generally recognized as the metastatic pathway from the liver to the mediastinal lymph nodes [33, 34], the details have not yet been elucidated. However, it is unlikely that liver tumors deep in the liver metastasize via this pathway. We have identified in this study the hepatic venous lymphatic system may be one of the important pathways. Although further detailed studies are needed, we consider the following lymphatic pathways. One is the pathway the lymphatic vessels around the hepatic vein continuous to IVC to the mediastinum via the diaphragm, and the other is the pathway from the inferior vena cava directly to the mediastinum. Hepatic tumors located deep in the liver may metastasize to mediastinal lymph nodes via these pathways. The hepatic venous lymphovascular system we have described could act as a metastatic pathway in such cases.

This study had the following limitations to be elucidated in the future. First, we investigated a small number of cadavers. However, the aims of this study were to elucidate the anatomical distribution and lymph dynamics in human liver for the first time. Therefore, this study was conducted with a limited sample size. We believe that further experiments with larger sample sizes are needed. Second, lymphatic vessels might have been injured during the whole-mount preparation of hepatic veins. Third, we did not study the entire lymphatic system of the liver. We revealed the lymphatic system around the major hepatic veins in this study. However, we did not reveal the heterogeneity among right, middle, or left hepatic veins. Furthermore, the lymphatic system around caudate lobe was not investigated. Forth, since this study was conducted on cadavers, future studies on lymphodynamics in vivo using Indocyanine green and other techniques are needed.

## Conclusions

We revealed the three-dimensional structure and lymphatic dynamics of the hepatic venous lymphatic network in humans. The hepatic venous lymphatic network is one of the most important intrahepatic lymphatic pathways and a clearer understanding may help to elucidate intrahepatic lymphatic physiology and pathologies, such as the spread of malignant disease.

## Acknowledgments

The authors thank Drs Yoshiya Asano, Daisuke Okano, Erina Saito, and Kazuto Takahashi for their assistance with generating tissue sections, tissue staining and immunohistochemistry.

## Author Contributions

**Conceptualization:** Kotaro Umemura, Hiroshi Shimoda, Kenichi Hakamada.

**Formal analysis:** Kotaro Umemura, Kenichi Hakamada.

**Investigation:** Kotaro Umemura, Hiroshi Shimoda, Keinosuke Ishido, Kentaro Sato, Yuto Mitsuhashi, Seiji Watanabe, Hirokazu Narita, Tomohiro Chiba.

**Methodology:** Kotaro Umemura, Hiroshi Shimoda, Norihisa Kimura, Taiichi Wakiya, Takuji Kagiya, Kentaro Sato, Kenichi Hakamada.

**Resources:** Hiroshi Shimoda.

**Supervision:** Hiroshi Shimoda, Keinosuke Ishido, Kenichi Hakamada.

**Writing – original draft:** Kotaro Umemura, Keinosuke Ishido, Norihisa Kimura, Kenichi Hakamada.

**Writing – review & editing:** Kotaro Umemura, Hiroshi Shimoda, Keinosuke Ishido, Kenichi Hakamada.

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
