## [Decision Letter · Decision Letter 0]

27 Mar 2023

PONE-D-23-03219Microanatomical Organization of Hepatic Venous Lymphatic System in HumansPLOS ONE

Dear Dr. Umemura,

Thank you for submitting your manuscript to PLOS ONE. After careful consideration, we feel that it has merit but does not fully meet PLOS ONE’s publication criteria as it currently stands. Therefore, we invite you to submit a revised version of the manuscript that addresses the points raised during the review process.

We look forward to receiving your revised manuscript.

Kind regards,

Shozo Yokoyama, M.D.,Ph.D.

Academic Editor

PLOS ONE

Journal Requirements:

"NO authors have competing interests

Enter: The authors have declared that no competing interests exist."

Reviewers' comments:

Reviewer's Responses to Questions

**Comments to the Author**

1. Is the manuscript technically sound, and do the data support the conclusions?

Reviewer #1: Partly

Reviewer #2: Yes

Reviewer #3: Yes

2. Has the statistical analysis been performed appropriately and rigorously? 

Reviewer #1: Yes

Reviewer #2: Yes

Reviewer #3: Yes

3. Have the authors made all data underlying the findings in their manuscript fully available?

Reviewer #1: Yes

Reviewer #2: Yes

Reviewer #3: No

4. Is the manuscript presented in an intelligible fashion and written in standard English?

Reviewer #1: Yes

Reviewer #2: Yes

Reviewer #3: Yes

5. Review Comments to the Author

Reviewer #1: Reviewer comments to PONE-D-23-03219

This study researched microanatomical organization of hepatic venous lymphatic system using three female human cadavers. Based on imaging results, the authors demonstrated lymphovascular networks form the central vein to the inferior vena cava.

As the authors mentioned, although the periportal lymphatic system was well discussed so far, the hepatic venous lymphatic system was not fully discussed. In this regard, this study had several novelties.

The reviewer had following comments.

Major comments:

1. In the discussion, the authors discussed about the connection of hepatic venous lymphatic systems to mediastinal lymph nodes. As the authors mentioned, the superficial lymphatic system is a possible pathway but not be elucidated. As for this study, as this study also did not show the connection to the mediastinal lymph nodes microscopically, authors’ discussion may not yet be elucidated.

2. Regarding the distribution of hepatic venous lymphatic systems, was there any heterogeneity among right, middle, or left hepatic veins? How about the hepatic venous lymphatic system in the caudate lobe? The reviewer found an article discussing about the heterogeneity of lymphatic drainage patterns among liver segments, although it was a study using a murine model (Frenkel et al. Sci Rep 2020:10:21808. doi: 10.1038/s41598-020-78727-y.). Please discuss about it.

Minor comments:

1. Please indicate the pathological findings of the liver tissue. Were these cadavers free from any liver disease such as hepatitis or fatty liver?

Reviewer #2: Dear editor I thank you for your invitation to assess the scientific quality of the paper titled: “Microanatomical Organization of Hepatic Venous Lymphatic System in Humans”.

The study reports a very impressive topic, micro-lymphatic vasculature of the liver by using different technique including immunohistochemistry which is very interesting.

The paper in very good in organization and presentation of the data. I kindly request the authors to respond for the following concerns.

1. Why the authors didn’t determine the sample size scientifically? Why only three cadavers were used? Is it possible to conclude on the 3D of hepatic lymphatic vasculature from the data collected from only three cadavers? Could the result be valid?

2. Why only female cadavers were used? Was it not possible to include male in order to compare if sexual dimorphism exist?

3. For how long the cadavers were preserved? What was the preservative used for embalming the cadaver? How the authors check whether the preservative affects the microcirculation of the liver? Since lymphatic wall is very thin.

4. The clinical application of such investigation should be better elaborate at the beginning of the discussion, so that it catches the attention of the readers.

5. In the discussion part, for your finding that states the presence of lymph vessels around thick walled veins than thinned ones, it is good to use development of lymph vessels to explain this.

Reviewer #3: Dr. Umemura and co-authors have written a manuscript on the topic of the anatomical structure of the hepatic venous lymphatic system in humans. The manuscript is well written and provides interesting insights into the lymphatic physiology of the human liver. The authors have conducted a post-mortem study of three human cadavers using both light microscopy, stereomicroscopy and scanning electron microscopy. The main finding is that lymphatic vessels are found in the adventitia of the sublobular veins with a wall thickness greater than 110 μM with an increasingly dense distribution in the proximal trunks of the hepatic veins, creating a rich lymphatic vascular network in the inferior vena cava.

I find the manuscript of good quality, with a clear description of methods and findings, proper use of statistical methods and conclusions that soundly supported by the observations. I only have a few comments that are listed below:

1) I think it would be relevant to elaborate a little on the human cadavers included in the study.

a. Was an informed consent obtained from the relatives of the deceased?

b. What were the causes of death?

c. How long time passed from the time of death to the specimen collection?

2) Did the authors observe any anatomical differences between the different subjects – e.g., with regards to the distribution density of lymphatic vessels? Would it be possible to plot the observations from the individual subjects with difference symbols/colours in the plots?

3) A minor point: I would suggest moving the section on statistics to the end of the methods section.

6. PLOS authors have the option to publish the peer review history of their article (what does this mean?). If published, this will include your full peer review and any attached files.

Reviewer #1: No

Reviewer #2: **Yes: **Melese Shenkut Abebe

Reviewer #3: No

---

## [Author Response · Author response to Decision Letter 0]

4 May 2023

Reviewer #1

In the discussion, the authors discussed about the connection of hepatic venous lymphatic systems to mediastinal lymph nodes. As the authors mentioned, the superficial lymphatic system is a possible pathway but not be elucidated. As for this study, as this study also did not show the connection to the mediastinal lymph nodes microscopically, authors’ discussion may not yet be elucidated

Reply)

We thank you for your suggestion. Per your remark, our study is not elucidated on that point. The superficial lymphatic system was reported as the metastatic pathway from the liver to the mediastinum (33,34). However, it is unlikely that liver tumors deep in the liver metastasize via this pathway. We have identified in this study the hepatic venous lymphatic system may be one of the important pathways. Although further detailed studies are needed, we assume the following lymphatic pathways. One is the pathway the lymphatic vessels around the hepatic vein continuous to IVC to the mediastinum via the diaphragm, and the other is the pathway from the inferior vena cava directly to the mediastinum. We have added the following text to the Discussion. 

However, it is unlikely that liver tumors deep in the liver metastasize via this pathway. We have identified in this study the hepatic venous lymphatic system may be one of the important pathways. Although further detailed studies are needed, we consider the following lymphatic pathways. One is the pathway the lymphatic vessels around the hepatic vein continuous to IVC to the mediastinum via the diaphragm, and the other is the pathway from the inferior vena cava directly to the mediastinum. Hepatic tumors located deep in the liver may metastasize to mediastinal lymph nodes via these pathways. Page 17, 379-386

Regarding the distribution of hepatic venous lymphatic systems, was there any heterogeneity among right, middle, or left hepatic veins? How about the hepatic venous lymphatic system in the caudate lobe? The reviewer found an article discussing about the heterogeneity of lymphatic drainage patterns among liver segments, although it was a study using a murine model (Frenkel et al. Sci Rep 2020:10:21808. doi: 10.1038/s41598-020-78727-y.). Please discuss about it.

Reply)

Thank you for your suggestion. In the literature you taught us, Frenkel et al. reported lymphatic drainage patterns in the murine liver in vivo. They included results similar to and different from ours. Although we studied the correlation between the diameters of hepatic veins and lymphatic vessel distribution, we did not elucidate differences in the distribution rate of lymphatic vessels in the right, middle, and left hepatic veins. We also did not examine the distribution rate of lymphatic vessels around the caudate lobe. We have cited their report in the paper because it was extremely important. Additionally, we added the following text to the Discussion. 

1.Frenkel et al. reported the drainage patterns involving a pathway from the inferior vena cava to mediastinal lymph nodes in the murine livers in vivo (28). They showed that much of the lymphatic fluid from the middle lobe of the liver flows to the peri-esophageal lymph nodes via the lymphatic vessels around the IVC. They also showed that lymphatic fluid from the right and left lobes of the liver flows to the hilar lymph node. There may be heterogeneity in lymphatic drainage patterns for each region in human liver. Page 16, 362-368

2. Third, we did not study the entire lymphatic system of the liver. We revealed the lymphatic system around the major hepatic veins in this study. However, we did not reveal the heterogeneity among right, middle, or left hepatic veins. Furthermore, the lymphatic system around caudate lobe was not investigated. Page 18 393-397

1. Please indicate the pathological findings of the liver tissue. Were these cadavers free from any liver disease such as hepatitis or fatty liver?

Reply)

We appreciate your remark. The pathological findings of the liver for each cadaver were as follows.

The pathological findings of Cadaver 1 showed mild hepatitis with mild inflammatory cell infiltration in the portal area, but no liver fibrosis or fatty liver.

The pathological findings of Cadaver 2 showed moderate hepatitis with fibrotic enlargement and moderate inflammatory cell infiltration in the portal area, but no fatty liver. The pathological findings on cadaver 3 did not show any liver disease, including fibrosis or hepatitis. Table 1 was added as details of the cadavers.s

Reviewer #2:

1. Why the authors didn’t determine the sample size scientifically? Why only three cadavers were used? Is it possible to conclude on the 3D of hepatic lymphatic vasculature from the data collected from only three cadavers? Could the result be valid?

Reply)

We appreciate your remarks. We conducted this study on three cadavers. The main studies on the three cadavers were as follows. Cadaver No.1: Sections across the hepatic veins and inferior vena cava were prepared and studied to clarify the lymphatic anatomy around the hepatic veins, including the sublobular vein. Cadaver No.2: To elucidate the lymph dynamics around the hepatic veins, sections were prepared from liver specimens in which Indian ink was injected into the right lobe and examined using microscope. Cadaver No.3: To clarify the three-dimensional structure and distribution of lymphatic vessels around hepatic veins, whole mount immunostaining was performed on a block of hepatic vein with the liver parenchyma removed and examined using stereomicroscope. We added to Table 1 as the details of cadavers including the purpose of the experiment.

Per your remark, the sample size was very small in this study. However, the aims of this study were to elucidate the anatomical distribution and lymph dynamics in human liver for the first time. Therefore, this study was conducted with a limited sample size. We believe that further experiments with larger sample sizes are needed. We added the following text to the Discussion.

First, we investigated a small number of cadavers. The sample size was very small in this study. However, the aims of this study were to elucidate the anatomical distribution and lymph dynamics in human liver for the first time. Therefore, this study was conducted with a limited sample size. We believe that further experiments with larger sample sizes are needed. Page 18, 388-392

2. Why only female cadavers were used? Was it not possible to include male in order to compare if sexual dimorphism exist?

Reply)

We appreciate your remark. I checked the gender of the three cadavers again. As a result, cadaver 1 was male, not female. We added Table1 to the Materials and Methods.

Page 5,6 97-109

3. For how long the cadavers were preserved? What was the preservative used for embalming the cadaver? How the authors check whether the preservative affects the microcirculation of the liver? Since lymphatic wall is very thin.

Reply)

We appreciate your remark. Preservation time were listed in Table 1. Postmortem, 10% formalin and 5% phenol were injected through the femoral artery of the cadavers and fixed for at least 1 week. After fixation, it was replaced with 60% alcohol + 5% phenol. Per your remark, Lymphatic vessels were observed collapsed in the 5 μm sections. However, the presence or absence of lymphatic vessels was not considered to be affected. We think further studies on microcirculation using ICG and other dyes in vivo are needed. We added the following text to the Materials and Methods and the Discussion.

Postmortem, 10% formalin and 5% phenol were injected through the femoral artery of the cadavers and fixed. They had no hepatic tumors or abdominal malformations. The details of the cadavers used in the experiment were shown in Table 1. Page5, 6 97-109

Forth, since this study was conducted on cadavers, future studies on lymphodynamics in vivo using ICG and other techniques are needed. Page 18 397-399

4. The clinical application of such investigation should be better elaborate at the beginning of the discussion, so that it catches the attention of the readers.

Reply) Thank you for your suggestion. Per your remark, we added the following text at the beginning of the Discussion.

Liver tumors, including metastatic liver tumors and primary liver cancer, are known to metastasize to mediastinal lymph nodes. The prognosis for patients with mediastinal lymph node metastases is extremely poor, and little is known about the pathway of metastasis. (17,18) Therefore, clarification of intrahepatic lymphatic anatomy is needed to elucidate the pathway of metastasis. We focused on the lymphatic anatomy around the hepatic veins. Page 13, 14, 291-296

5. In the discussion part, for your finding that states the presence of lymph vessels around thick walled veins than thinned ones, it is good to use development of lymph vessels to explain this.

Reply)

We appreciate your remark. We added the following sentence in discussion. 

This may suggest that the lymphatic vessels around the hepatic veins originate from the adventitia of the sublobular veins and develop more densely from the hepatic veins to the inferior vena cava (Fig 7). Page 14, 314-316

Reviewer #3

1) I think it would be relevant to elaborate a little on the human cadavers included in the study.

a. Was an informed consent obtained from the relatives of the deceased?

Reply)

Thank you for your suggestion. We obtained consent from the individuals and their relatives before their death to be a specimen. The donors have been enrolled before their death in an association of people who have offered to donate their bodies with the consent of themselves and their relatives, provided that they are not infected with human immunodeficiency virus, hepatitis B virus, or hepatitis C virus. Additionally, we received opt-out consent from the relatives for this study. We added the following text to the Materials and Methods.

The donors were enrolled before their death in an association of people who had offered to donate their bodies with the consent of themselves and their relatives, provided that they were not infected with human immunodeficiency virus, hepatitis B virus, or hepatitis C virus. Additionally, we received opt-out consent from the relatives for this study. Page5 88-93, 95

b. What were the causes of death?

Reply) Thank you for your remark. We added the details of the cadavers, including cause of death, in Table 1.

c. How long time passed from the time of death to the specimen collection?

Reply) Thank you for your remark. We added the details of the cadavers, including the time before specimen collection, in Table 1.

2) Did the authors observe any anatomical differences between the different subjects – e.g., with regards to the distribution density of lymphatic vessels? Would it be possible to plot the observations from the individual subjects with difference symbols/colours in the plots?

Reply) We appreciate your remarks. Unfortunately, only one cadaver (No.3) was measured and used as statistical data in this study, so it is not possible to show differences among the individuals. It is necessary to increase the number of cadavers and examine anatomical differences in the future. We added the following text to the Results and the Discussion.

Microscopically, dense lymphatic vessels were observed in the more central part of the hepatic veins. Therefore, we examined the relationship between the diameter of hepatic veins and the density of lymphatic vessels. Page11 234-236

However, the aims of this study were to elucidate the anatomical distribution and lymph dynamics in human liver for the first time. Therefore, this study was conducted with a limited sample size. We believe that further experiments with larger sample sizes are needed. Page18 389-392

3) A minor point: I would suggest moving the section on statistics to the end of the methods section.

Reply)

We appreciate your remark. We have made the changes as you indicated. Page 9, 10 186-195

---

## [Decision Letter · Decision Letter 1]

15 May 2023

Microanatomical Organization of Hepatic Venous Lymphatic System in  Humans

PONE-D-23-03219R1

Dear Dr. Umemura,

We’re pleased to inform you that your manuscript has been judged scientifically suitable for publication and will be formally accepted for publication once it meets all outstanding technical requirements.

Kind regards,

Shozo Yokoyama, M.D.,Ph.D.

Academic Editor

PLOS ONE

Reviewers' comments:

Reviewer's Responses to Questions

**Comments to the Author**

1. If the authors have adequately addressed your comments raised in a previous round of review and you feel that this manuscript is now acceptable for publication, you may indicate that here to bypass the “Comments to the Author” section, enter your conflict of interest statement in the “Confidential to Editor” section, and submit your "Accept" recommendation.

Reviewer #1: All comments have been addressed

Reviewer #2: All comments have been addressed

Reviewer #3: All comments have been addressed

2. Is the manuscript technically sound, and do the data support the conclusions?

Reviewer #1: Yes

Reviewer #2: Yes

Reviewer #3: Yes

3. Has the statistical analysis been performed appropriately and rigorously? 

Reviewer #1: Yes

Reviewer #2: Yes

Reviewer #3: Yes

4. Have the authors made all data underlying the findings in their manuscript fully available?

Reviewer #1: Yes

Reviewer #2: Yes

Reviewer #3: Yes

5. Is the manuscript presented in an intelligible fashion and written in standard English?

Reviewer #1: Yes

Reviewer #2: Yes

Reviewer #3: Yes

6. Review Comments to the Author

Reviewer #1: (No Response)

Reviewer #2: I would like to thank the authors.

They responded for all my comments and I am happy with that.

Best of luck.

Reviewer #3: No further comments

(..............................................................................)

7. PLOS authors have the option to publish the peer review history of their article (what does this mean?). If published, this will include your full peer review and any attached files.

Reviewer #1: No

Reviewer #2: No

Reviewer #3: No

---

## [Editor Report · Acceptance letter]

17 May 2023

PONE-D-23-03219R1 

Microanatomical Organization of Hepatic Venous Lymphatic System in  Humans 

Dear Dr. Umemura:

I'm pleased to inform you that your manuscript has been deemed suitable for publication in PLOS ONE. Congratulations! Your manuscript is now with our production department. 

Kind regards, 

on behalf of

Dr. Shozo Yokoyama 

Academic Editor

PLOS ONE